# Implementing Patient Falls Education in Hospitals: A Mixed-Methods Trial

**DOI:** 10.3390/healthcare10071298

**Published:** 2022-07-13

**Authors:** Hazel Heng, Debra Kiegaldie, Louise Shaw, Dana Jazayeri, Anne-Marie Hill, Meg E. Morris

**Affiliations:** 1School of Allied Health, Human Services and Sport, La Trobe University, Melbourne, VIC 3086, Australia; h.heng@latrobe.edu.au (H.H.); louise.shaw@holmesglen.edu.au (L.S.); dana.jazayeri@unimelb.edu.au (D.J.); 2Northern Health, Melbourne, VIC 3076, Australia; 3Holmesglen Institute and Healthscope, Monash University, Melbourne, VIC 3800, Australia; debra.kiegaldie@holmesglen.edu.au; 4The Victorian Rehabilitation Centre, Healthscope, ARCH La Trobe University, Melbourne, VIC 3086, Australia; 5Western Australian Centre for Health & Ageing, School of Allied Health, The University of Western Australia, Perth, WA 6009, Australia; anne-marie.hill@uwa.edu.au

**Keywords:** falls, falls prevention, injury prevention, patient education, hospital, healthcare, accidental falls

## Abstract

Patient education is key to preventing hospital falls yet is inconsistently implemented by health professionals. A mixed methods study was conducted involving a ward-based evaluation of patients receiving education from health professionals using a scripted conversation guide with a falls prevention brochure, followed by semi-structured qualitative interviews with a purposive sample of health professionals involved in delivering the intervention. Over five weeks, 37 patients consented to surveys (intervention *n* = 27; control *n* = 10). The quantitative evaluation showed that falls prevention education was not systematically implemented in the trial ward. Seven individual interviews were conducted with health professionals to understand the reasons why implementation failed. Perceived barriers included time constraints, limited interprofessional collaboration, and a lack of staff input into designing the research project and patient interventions. Perceived enablers included support from senior staff, consistent reinforcement of falls education by health professionals, and fostering patient empowerment and engagement. Recommended strategies to enhance implementation included ensuring processes were in place supporting health professional accountability, the inclusion of stakeholder input in designing the falls intervention and implementation processes, as well as leadership engagement in falls prevention education. Although health professionals play a key role in delivering evidence-based falls prevention education in hospitals, implementation can be compromised by staff capacity, capability, and opportunities for co-design with patients and researchers. Organisational buy-in to practice change facilitates the implementation of evidence-based falls prevention activities.

## 1. Introduction

A key role of health professionals who work in hospitals is to educate patients about falls prevention [1]. Patient education is an essential part of falls prevention, given that up to 20–30% of hospital incidents are falls [2]. Falls prevention education aims to improve patient knowledge about falls and falls risks, and to teach patients and their carers about falls mitigation [3]. By educating patients, health professionals can help to minimise the mismatch between perceived risk and actual risk of falling whilst in hospital [4].

While recent studies highlight the potential of patient education to reduce falls, most trials have employed research staff to deliver the intervention rather than health professionals [2,5,6]. These trials were often carried out in sub-acute or rehabilitation wards, with few studies focusing on the acute setting [5]. In hospitals, patient education is usually delivered by nurses and allied health professionals [7,8]. Despite the evidence showing that educating patients can reduce hospital falls [9,10], nurses and allied health professionals arguably face barriers to delivering patient education in everyday practice [11]. Time constraints mean that patient education is not always implemented early. As there is a higher risk of falls in the first week of a hospital admission [12,13], it is important for health professionals to prioritise the delivery of evidence-based education very soon after admission.

One strategy to assist clinicians to implement evidence-based practice is to provide them with a guide for delivering falls education in the form of a scripted conversation. Scripted conversation guides have been used to educate patients in different healthcare contexts, such as advanced care planning [14], pre-operative care [15], and post-discharge falls prevention [16]. Scripted guides provide clinicians with a standardized and consistent method of delivering education whilst still tailoring the content to individual needs. This approach is favoured by hospital patients [11]. Another strategy is to provide patients with evidence-based falls prevention brochures [9,17]. The advantage of brochures is that patients can access falls prevention information at any time and as many times as needed. A combination of education modes has been shown to be more effective than relying on a single modality for delivering falls prevention education [10,17]. Combining a scripted conversation delivered by a health professional coupled with a patient fall prevention brochure, is arguably a robust method for delivering falls prevention education in hospitals.

Despite the evidence showing the benefits of patient falls education [2,17], the implementation of evidence into routine hospital practice to prevent falls and associated injuries remains challenging [18,19,20]. There can be barriers to translating evidence into practice such as cost, inconvenience, lack of confidence and skills, and social and environmental context [18,21]. While there have been some studies investigating falls research implementation barriers [22,23,24], few have done so in the context of hospital falls education research.

We conducted a mixed-methods study that aimed to compare usual care with health professional delivery of an interactive patient falls education intervention using a scripted conversation coupled with an evidence-based patient education brochure. Usual care included a hospital-wide policy on falls prevention strategies such as safe footwear, physical activity, assistive devices, environmental modifications, management of cognitive impairment, diet and medication reviews, and clinician and patient education that was informal and unstructured. We also aimed to identify enablers, barriers, and strategies to improve the implementation of falls education delivered by nurses, allied health professionals, and other hospital staff.

## 2. Materials and Methods

### 2.1. Study Design

This was a mixed methods study involving a quantitative analysis of a hospital-based patient education intervention with follow up semi-structured qualitative interviews of health professionals involved in delivering the intervention. This design was chosen as the qualitative phase would help to interpret and build upon the results of the quantitative data [25]. For the quantitative analysis, patients were randomised to an experimental group or control group. We used a computer random number generator to assign the hospital wards to receive either usual care or additional falls prevention education. Health professionals who worked on the experimental group ward delivered the intervention. They received education and training prior to implementation [26]. The control ward patients received usual care. Health professionals working in the control ward did not receive any additional education or training. It was not possible to blind the research assessors; however, the researchers were not involved in supervising health professionals or the care of patients, thereby reducing risk of bias. There were no associations between the participants and the research team.

For the qualitative interviews, a purposive sample of staff members who were involved in coordinating and delivering the patient education were interviewed to investigate barriers, enablers, and recommended strategies for implementation.

### 2.2. Ethical Considerations

This research was supported by an Australian NHMRC partnership grant (GNT1152853) and was registered on the Australian New Zealand Clinical Trials Registry (ACTRN12621000430831). Ethics approval was granted by La Trobe University (HEC21023).

### 2.3. Study Population and Setting

Participants for the quantitative analysis were patients in either a medical or surgical ward of a private Australian acute care hospital. The wards were randomised to be either a control ward (usual care) or an intervention ward (receiving a scripted conversation about falls prevention and a brochure). Recruitment occurred over a 5-week period in 2021. Patients were excluded from the study if they were not medically stable, had cognitive impairment, or were not able to communicate in English.

For the qualitative study, nurses and allied health professionals were eligible to participate if they attended the health professional education training session and delivered the education intervention. Managers were recruited if they were involved with the study, such as assisting with coordinating the trial processes. Recruitment occurred 2 months after the end of the quantitative component. This delay was due to limited availability of staff, awaiting ethics amendment approval, and staff shortages associated with the COVID-19 pandemic.

### 2.4. Intervention

The falls prevention intervention was delivered by nurses and allied health professionals and consisted of a face-to-face scripted conversation about how to prevent falling and a standardized evidence-based hospital falls prevention brochure in addition to usual care. The conversation was based on the principles of the Safe Recovery Program [6,27,28] and was simplified to focus on clarifying falls prevention information, teach-back methods, and goal setting. The intervention aimed to empower patients to take responsibility for their own fall prevention, while being feasible for ward staff to deliver given existing time constraints. It followed the quality metric tool used in Heng et al. [17], which was adapted from the 4P model of educational principles. The learner and teacher characteristics, learning activities, and outcome evaluation were considered. The intervention was designed to be delivered within 24 h of patient admission to the ward and included having follow up conversations with the patient to reinforce education and check on attainment of goals previously set by the patient. The intervention underwent several iterations following feedback obtained from a panel of clinicians, experts in falls prevention, and consumers.

One week prior to implementation, nurses and allied health professionals attended a 1 h training session in person or through videoconferencing. The training was provided by a researcher (LS) who was an experienced clinical educator. The training session included provision of the latest evidence on patient education for falls prevention, pre-recorded vignettes of simulated delivery of the intervention, and small group discussion. The full description of the training can be found in Shaw et al. [26]. A training session was recorded for staff to access online if they were not able to attend in person. Consent was gained at this stage for a follow-up interview with each health professional.

### 2.5. Data Collection

Quantitative data were collected from patients using a survey that included demographic data and self-reported responses to seven items on a five-point Likert scale. The survey was based on an instrument by Khong et al. [29] and evaluated patient knowledge about falls, self-perceived risk of falls, falls behaviour change, as well as patient views on the learning experience (Appendix A). Eligible patients completed the pre-test survey at the bedside with a researcher (H.H. or R.L.) once written consent was provided. The same researchers were responsible for collecting post-test data 1–2 days after participants received the intervention. The same procedure was carried out in the control ward.

For consenting health professionals, individual semi-structured telephone interviews were conducted by a researcher (D.K.) at their convenience. Telephone interviews were necessary due to the COVID-19 pandemic restrictions at the time [30]. Two separate interview schedules were designed after expert advice and discussions between research team members and experienced qualitative researchers. One schedule was for health professionals who were directly involved with patient education (Appendix A), while the other schedule was for health professional managers who did not deliver the intervention but were associated with the implementation process of the first phase (Appendix A). An independent transcription service transcribed each interview. Participants were de-identified and assigned a number in each transcript.

### 2.6. Data Analysis

For the quantitative phase, distributions and measures of central tendency were used to describe the demographic make-up of each patient group. Patient data were subjected to descriptive analysis where appropriate.

A qualitative descriptive approach with thematic analysis was used to identify themes and sub-themes from the interviews [31,32,33]. For the qualitative data, template analysis using a coding template to represent themes from the data was used to identify patterns and ideas from the interview transcripts [34,35]. The main themes pertained to perceptions and experiences of participants in the context of the research questions and interview schedules [31] Two researchers (H.H., L.S.) independently analysed and coded the qualitative interview data. The interviews were focused on barriers, enablers, and recommended strategies for implementation of patient falls prevention education. Findings were discussed via video-conferencing meetings. A third researcher (D.K.) was consulted to achieve consensus if there were differences in opinion. Coded data were presented in spreadsheets and tables. Quotes were chosen to support final themes and sub-themes.

## 3. Results

### 3.1. Quantitative Data

Over five weeks, 122 hospital patients were approached with 85 declining to consent as they had no interest in participating in the study or were planned for discharge the following day. In total, demographic data and pre-test surveys were collected from 37 participants (intervention *n* = 27; control *n* = 10). The full details of patient characteristics can be found in Table 1. There were 16 women in the patient intervention group (59.3%) and 5 women in the control group (50.0%). The mean age of the intervention group was 15 years greater than the control group (intervention 75.3 SD 16.56, range 36–96; control 60.2 SD 22.26, range 28–94).

In the intervention group, the mean proportion of patients who received the intervention on the ward each day was 15.3% (0.153 SD 0.06, range 0.074–0.267). On average, 22.1% of patients on the ward each day were reported to be cognitively impaired as per ward records (0.221 SD 0.05, range 0.133–0.308). For the 27 patients in the intervention group, 25 provided pre-test data and only 3 provided post-test data. One consented to providing pre- and post-test data while two completed the post-test survey without completing the pre-test survey beforehand. Within the control group, post-test data were collected from eight participants with an average of 1.625 days (SD 0.7) between pre- and post-test surveys. Due to the inability to obtain sufficient paired responses (pre and post), inferential statistics could not be calculated. Throughout the trial period, which occurred during the COVID-19 pandemic, the research team implemented several strategies to improve intervention delivery and staff engagement in the project and data collection. This included increasing the presence of researchers on the ward to support staff and respond to questions, engaging senior leadership, reminders during handover, and input from executive staff. The average number of interventions carried out doubled, however, this was not sustained past week four.

### 3.2. Qualitative Analysis

Seven individual interviews were conducted with health professionals over three weeks. This was the full complement of consenting hospital staff. The participants were enrolled nurses (*n* = 3), a registered nurse (*n* = 1), physiotherapist (*n* = 1), assistant nurse unit manager (*n* = 1), and an executive staff member with a background in nursing (*n* = 1). The average duration of the interviews was 20.6 min (SD 4.97).

From an analysis of views expressed by the interviewed participants, three broad themes about implementation emerged (Table 2): (i) barriers to implementing falls education; (ii) enablers for implementation of falls education; and (iii) recommended strategies to improve implementation. Quotes supporting each theme has been included in Appendix A.

### 3.3. Barriers to Implementing Falls Education

Several interviewees indicated that the patient status affected patient engagement, with a significant proportion of patients on the intervention ward having cognitive impairment or being too medically unwell to participate. Many of them felt that they were not able to deliver the falls education intervention if the patient was cognitively impaired, and unable to understand the information. Others noted that some patients had language barriers or did not recognise their risk of falls and therefore declined to engage with the clinician.

“…the biggest barrier was probably the cognitive impaired patients, anyone with a language barrier”(P5, enrolled nurse.)

Staff attitudes were also recognised by participants as contributing to reduced engagement with the implementation process such as previous unsuccessful falls prevention measures, patient-related barriers, or workload demands. As a result, there was a perception that staff were reluctant to carry out the intervention. A lack of confidence or knowledge was also acknowledged as being a barrier.

“Like, some felt like it’s just a time-consuming thing without real benefit for our class of patients”.(P2, registered nurse.)

“the nursing staff’s knowledge base of that particular subject would determine how successful, and obviously, their ability to engage with a different level and different variety of patients, really, depending on their condition…”(P6, executive.)

Despite reporting strong interprofessional teamwork prior to the trial, interviewees acknowledged that there was limited collaboration in relation to falls education. Intra- (within profession) and interprofessional (between professions) communication and collaboration were encouraged during the clinician training, however, participants noted that there was no specific discussion among staff during the trial. Staff may therefore not have felt the need to focus on further collaboration due to the pre-existing interprofessional relationship.

“…overall on the ward, there might be—possibly like there might not be the best sort of communication across all the other disciplines, and that’s where you might run into trouble”.(P1, physiotherapist.)

Some commented that there was a lack of staff input into designing the intervention and the trial process, which may have contributed to staff being unsure about the expectations and procedures. Some of the interview participants felt that if staff were consulted beforehand, they could have highlighted issues with ward processes or be better prepared in how to appropriately implement the patient education. These issues could have been remedied prior to the start of the trial, thus improving implementation outcomes.

“There wasn’t any consultation with the staff of how would be the best way to deliver this, and no real conversation regarding what were the expectations of the staff going into this”.(P3, enrolled nurse.)

The modelling scenarios used in the training session were thought to be helpful by those interviewed but they felt it did not fully encompass realistic settings. They perceived that the scenarios did not accurately reflect barriers such as language barriers and time constraints, which resulted in creating challenges when attempting to adapt the scripted conversation to these situations.

“…in a real-life situation when the nurse is working…the dynamics change. You know when you’re watching [the training]…you’re doing this thing, and that’s what you have to do. Where, in a realistic, real-life work situation where you have five patients to look after, this one is buzzing, maybe this one need toileting… In the middle of an education program someone can buzz, …there can be an emergency call in another room, and the alarms are going off. You need to attend to that. You need to end your education pathway to [work through it]. So it wasn’t really practical in the real-life situation”.(P2, registered nurse.)

Other barriers reported included time constraints, a perception by some staff that there was limited leadership support and organisational management underpinning patient education, and few processes to support implementation of the research evidence into clinical practice. Lack of staff time was cited as a major barrier to delivering patient falls education. Interviewees agreed that time was limited on the ward due to staffing ratios, level of care required for patients, and existing high workloads. Some participants reported not receiving enough information prior to the trial, which led to them feeling unsupported and at times frustrated. Others highlighted the importance of leadership engagement to improve ward staff commitment to the patient education process. Several of the staff participants identified process limitations such as a breakdown in top-down communication and uncertainty in the delegation of responsibilities that may have contributed to implementation difficulties.

“We know that during this period, from what I recall, the manager…had some leave…There was some probably not-ideal communication there, we could have probably tightened up on that. And so, there was some areas where our communication probably wasn’t as tight as it should have been, and probably some other issues going on there, personal issues”.(P6, executive.)

Inconsistencies in ward processes were also perceived to be an issue. For example, one nurse commented that the intervention relied on patients receiving falls prevention brochures at admission, however, this had ceased previously due to COVID-19 restrictions. This was perceived to have had an impact on the delivery of education to patients and their families.

“…there are directives that come down from the top that are made without any, what feels like, without much input from us on the bottom line”(P7, enrolled nurse.)

### 3.4. Enablers for Implementation of Falls Education

Fostering patient empowerment with falls prevention was one factor that motivated the interviewed staff to implement evidence-based falls education. They perceived that the scripted dialogue and falls brochure engaged, educated, and empowered patients to prevent falling. Additionally, they felt that the patient falls prevention education intervention involved patients in decision-making, which allowed patients to have some autonomy over the process. As a result, they felt patients were more likely to adhere to falls prevention strategies. This was helpful in building a positive relationship between the patient and staff. For the health professional participants, knowing that the trial was an evidence-based intervention was a motivating reason to implement the patient education despite existing barriers. They considered evidence-based practice to be an essential part of healthcare and trusted that the intervention would be effective in preventing falls.

“for me, being able to be involved in something like this, …putting into practice having evidence-based care, as a clinician that’s personally important. I think the development of evidence-based care is essential to nursing. I mean, it’s [the only way] we get anywhere really. So, it was good…knowing that that’s what it was”.(P3, enrolled nurse.)

Other views on enablers for implementation related to existing intra and interprofessional collaboration, consistent reinforcement of falls education, having a structured and embedded approach to falls prevention, and the use of modelling scenarios during training. The interview participants agreed that there was a strong existing collaborative relationship between nurses and allied health professionals. Examples were provided of allied health professionals engaging with nursing staff about falls prevention prior to the commencement of the trial. Participants also reported a strong intraprofessional (within profession) communication and collaboration.

“I didn’t have a lot of discussions about this program, but allied health is always discussing with us nurses about the mobility and falls risk and so forth of the patient”.(P5, enrolled nurse.)

Interviewees also found that having a consistency of falls education provision across and within professions enhanced implementation. Reinforcement of the same message from the healthcare team was cited as a key feature to encourage clinicians to implement the intervention.

“…we were all educated about it, around about the same time, so we…were on the same page, which I think was more or less a protective factor in terms of implementing the work”(P3, enrolled nurse.)

Formalising and structuring the intervention and having resources available was thought to be helpful in delivering the patient education. The resources perceived to be of value included the printed guides for staff and the recording of the training session. One nurse identified that incorporating the intervention into usual care helped with implementation. They also appreciated the scenarios that were presented during the training session. Watching the scenarios helped them understand how the intervention was to be delivered, how to respond to patients, and it developed their confidence in carrying out the scripted conversation.

“it was like a real-life scenario, hearing it actually out loud and sort of being role played, yeah, it showed how the conversation can flow and it showed how it can work”.(P7, enrolled nurse.)

A strong enabler for implementation of the intervention expressed by participants was the support from senior staff. Several participants reported that it was motivating when they received frequent reminders and encouragement from senior staff to implement the intervention.

“I know that when we started a shift we were reminded about the program, so that was very good to remind [us]”.(P5, enrolled nurse.)

### 3.5. Recommended Strategies to Improve Implementation of Patient Falls Education

Several recommendations were proposed by the staff interviewed to improve implementation outcomes. Ensuring accountability, both between staff and towards patients, was one strategy raised. Some felt that by being accountable for patients in their care, staff would naturally carry out interventions that were beneficial to patients. It was also suggested that staff be accountable amongst themselves by supporting each other with frequent check-ins and reminders.

“…it’s about knowing your patient and being accountable you start to know your patient. And then if you know your patient, then you start to sort of go to that next step and think about ways to improve their care”.(P1, physiotherapist.)

Another participant proposed focusing further on intra- and interprofessional collaboration about the intervention. Some examples given were to have discussions and specific conversations about the intervention during handover sessions and between groups. This would increase awareness among staff and improve implementation. Engaging staff and providing support was another strongly recommended strategy. Suggestions included involving staff in identifying barriers, advising staff how the intervention adds value to their work, informing staff about the positive changes arising from implementation of the intervention, and incentives. Some suggested increased support from research staff and further training and education for clinicians would help. A few interview participants felt that implementation outcomes would improve if the intervention was embedded into usual practice. By making it a part of their daily routine, staff would find it easier to carry out falls education. Initiating the intervention early and implementing it throughout the patient’s stay may assist as well.

“We have some standard protocols that needs to be done while the patient is getting admitted, so just as part of the admission process. I think that might make a habit of them to just do it”(P4, assistant nurse unit manager.)

In line with the barriers previously identified, some interviewees strongly believed that stakeholder input on the design of the intervention along with leadership engagement and support would greatly improve feasibility of delivery and implementation outcomes. Strong leadership engagement and support may also improve communication amongst ward staff and encourage uptake and adherence. Having hospital systems and policies aimed at supporting the implementation of patient education was another suggested strategy. Organising regular meetings and audits was also considered to be a helpful way of ensuring improved implementation processes.

“…clearly, once we’ve got the buy-in from the executive, and then there needs to be the buy-in from the manager to be on the same page, in terms of what we’re trying to achieve and what the actual issues are, and the involvement in the actual research project, and to really have that buy-in, that we need to give this a go. And then, of course, then it’s disseminated amongst the ANUMs [associate nurse unit managers] and then filtered down to the other staff”.(P6, executive.)

## 4. Discussion

Despite the strong body of literature showing that person-centred and interactive patient falls prevention education can be a powerful determinant of the frequency of hospital falls [2,9,17], the behaviour change associated with health professionals adapting their routine clinical practice to implement evidence-based patient education can be challenging to achieve [36,37,38]. Even though steps were taken in the current study to ensure effective delivery of the evidence-based falls prevention intervention, only a small proportion of hospital patients received the intervention as part of this trial. Running the trial during the COVID-19 pandemic was one factor that might have contributed to this issue.

Our experience of implementation challenges is not uncommon. Other studies have also investigated barriers to falls prevention implementation programs [22,23,24]. The themes raised in this study were similar to the findings of Ayton et al. [22], Semin-Goossens et al. [23], and O’Connell and Myers [24]. Of note, the importance of leadership engagement to achieve the implementation of evidence into practice was highlighted [22,23,24]. The qualitative data from health professionals in the current study provided valuable insights into the enablers and barriers of implementation of evidence-based falls education.

Time constraints and heavy workloads were a major barrier to health professionals delivering falls education within hospital wards in the current trial. This finding was in agreement with several studies in acute and sub-acute settings [39,40]. Health professionals often face a heavy workload, particularly in acute care hospital settings [41,42]. This may have been exacerbated by the COVID-19 pandemic in this current study. Even though falls prevention is talked about as a priority for most health professionals [8], a lack of time is consistently cited as a barrier to carrying out evidence-based practice [39,43]. For example, a study by McKenzie et al. [44] showed that the pressure of the work environment and heavy workloads were reported as a barrier in implementing an intervention for creating a safety culture. Carrying out the intervention was considered an additional task rather than being a core part of the role of health professionals. This was reflected in the qualitative data where a lack of time and increased workloads were reported to impact on intervention delivery.

Reduced engagement from some leaders was another perceived barrier to the implementation of evidence-informed patient education. While health professionals acknowledged support from senior staff as an enabler, they identified that further communication and support from leaders could have improved implementation outcomes. When senior members of an organisation are supportive of change, staff are more likely to carry out evidence-based interventions [39,43,44]. Yost et al. [45] noted that having supportive organisational leaders can lead to more successful outcomes. By engaging leaders, further barriers could be overcome, such as identifying pre-existing limitations in institutional processes. Similarly, including stakeholder input when designing patient falls education resources was a recommendation that arose from this trial. This is in agreement with O’Cathain et al. [46], which identifies stakeholder input as crucial when developing health interventions. A strong consumer engagement allows for a better designed trial and intervention that meets the needs of stakeholders, leading to more feasible interventions and effective implementation outcomes [47].

Several factors can affect the implementation of research into clinical practice [36,37]. Behaviour change has been shown by Michie et al. [48] to be key. They advised that individuals need to have capability, motivation, and access to opportunities for effective behaviour change. A theoretical domains framework can also be utilised to target specific domains within each component of the behaviour change wheel [48,49]. For example, to improve an individual’s reflective motivation, the intervention may need to address health professional beliefs about their capabilities and consequences, or their intentions and goals [49]. Applying these behaviour change theories will allow for a better design of implementation interventions [50]. In addition, educational models such as the 4P model of educational design [51], which emphasizes stakeholder engagement and management support, knowing the context of the learning environment, and having a full appreciation of learner and teacher characteristics prior to implementation, may also assist with implementation.

Some of the ways that health professionals can be supported to implement evidence into practice include training, education, and modelling from the leadership team. Our study determined that receiving this training can foster motivation to implement strategies to empower patients to prevent falling whilst in hospital. Creating an awareness about falls risk factors and preventative measures should be reinforced in health professional training to encourage implementation [22,28]. A recent study by Provvidenza et al. [52] highlighted that training delivered at an organisational level can be effective in improving knowledge translation in hospitals. The nature of knowledge translation and implementing research is known to be complex [20]. Promoting accountability among individuals as well as between the organisation and the workforce was a recommendation that was raised in our current study. This was in agreement with other studies conducted in the context of fostering a safety climate in hospitals [44,53,54]. Falls and associated injuries are a persistent safety challenge in hospitals globally [1,10].

There were limitations of the current study. Only patients without cognitive impairment were included because they needed to be able to read and respond to patient education pamphlets. For future trials, cognitively impaired patients could be included. Strategies such as repetition, rephrasing, and frequent positive reinforcement can be considered when delivering education to the cognitively impaired [17,55]. Training health professionals about these strategies by using role play or working with simulated patients can also be contemplated [56]. In addition, we did not quantify the effects of the intervention on falls risk, rates, or injuries. Only acute wards participated and sub-acute or rehabilitation wards may have different outcomes given the longer length of stay. The similarities and differences of the wards and how the context of the ward may have shaped interviewees insights were not analysed. Another limitation was the short timeframe for data collection. If the study could be conducted over a longer period, staff may have had more time to understand and follow implementation processes leading to more patient data. The COVID-19 pandemic was a major limitation to data collection. Due to the restrictions imposed by the pandemic, interviews with staff needed to be conducted by telephone and staff training sessions were modified to incorporate social distancing and personal protective equipment. Using personal protective equipment and masks may have impacted on program delivery because comprehension challenges could have been misidentified as a language or cognitive barrier.

## 5. Conclusions

Ensuring that health professionals consistently deliver evidence-based interventions in hospitals is an ongoing challenge. Time constraints, heavy workloads, and limited engagement from staff in leadership roles can sometimes be barriers to the rapid translation of research evidence into clinical practice. Supportive organisational leaders, engaging stakeholders in research, and promoting accountability among staff may lead to better implementation outcomes and enhance the effectiveness of patient falls prevention education.

## Figures and Tables

**Table 1 healthcare-10-01298-t001:** Patient participant characteristics.

Participant Characteristics	Control	Intervention
	(*n* = 10)	(*n* = 27)
Mean age, y (SD)	60.2 (22.26)	75.3 (16.56)
Gender		
Female	5	16
Male	5	11
Number of fallers (in past 12 months)		
1 fall	1	6
>1 fall	1	8
Number of co-morbidities +		
None	0	2
1–4	7	14
5–8	3	9
9–12	0	2
Reason for hospitalisation		
Medical diagnoses *	3	21
Orthopaedic	0	1
Respiratory	1	5
Other surgeries **	6	0

+ data from one participant missing. * includes digestive system disorders, infections, cancer, and renal disorders. ** other than orthopaedic surgeries.

**Table 2 healthcare-10-01298-t002:** Summary of themes.

Themes	Points Raised
Barriers	Patient status and/or impaired cognition. Staff attitudes to evidence implementation. Limited interprofessional collaboration in falls prevention. Lack of staff input in designing intervention and trial process. Unrealistic modelling scenarios in the staff training. Time constraints of staff. Sub-optimal leadership and organisational input. Limited processes to support implementation.
Enablers	Fostering patient empowerment and engagement with falls prevention. Existing intra- and interprofessional collaborations. Consistent reinforcement of falls education. Structured and embedded approach to falls prevention. Modelling scenarios during staff training. Support from senior staff.
Recommended strategies	Ensuring accountability between staff and towards patients. Focusing on intra- and inter-professional collaboration about the intervention. Stakeholder input into designing the intervention and implementation processes. Leadership engagement and support of systems to prevent hospital falls.

## Data Availability

The data presented in this study are available in Supplementary file D.

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
