# Peer review of "Implementing Patient Falls Education in Hospitals: A Mixed-Methods Trial"

_healthcare, 2022, doi:10.3390/healthcare10071298_

Round 1

Reviewer 1 Report

The paper presents an evaluation of the challenges health professionals find in delivering fall prevention information to patients in a hospital.

The paper presents a comprehensive analysis of the data gathered from interviews with health care professionals who participated in their study, highlighting barriers and enablers to patient education in hospital settings.

The study is clearly described, with relevant literature and supplementary material.

The study is centred on the attitudes and feedback from health professionals – main stakeholder in charge of patient education, rather than on patients at risk of falls.

The paper focus on challenges that are extrinsic to the programme/study, i.e. organisational buy in and staff accountability, which resulted in time constraints, lack of input and collaboration. Those challenges are relevant, authors present these as covered in the existing literature and need further studies to be addressed.

Authors need to clarify a number of points for a stronger contribution, as many of the barriers and enablers highlighted have been previously discussed (e.g. Francis-Coad et al 2020).

I hope authors find the following questions and comments helpful to position their work and improve the paper’s contribution:

Line 68 How the design of the present study took into account the challenges and issues reported in the literature? Please clarify the difference between the programme, study and analysis reported in the present paper and author’s positioning in the existing literature

85 Please add a comment on how healthcare professionals were recruited for this study.  Were they involved somehow, or consulted for the design of the study and intervention? Authors should comment how the design of the study as well as the design of the educational programme could have been improved to facilitate adherence from patients and staff, or include recommendations for future work.

86 Reference number 25 could not be found. If unpublished, please add details on the falls prevention programme delivered to the patients.

 123-126 Referenced papers highlighted the benefits and difficulties to ensure quality of falls prevention training for healthcare professionals (Heng et al 2021, and others). Please include more details on the content and format of the training session.

Table 1 Columns and headers should be re-organised to be easier to read.

Table 2 and discussion: Most of enablers and barriers presented are extrinsic to the programme. I would expect authors to include a self-reflection on what could have been done differently to facilitate access and adherence to the study. The “Points raised” (emerging themes from the interviews) could have been divided into categories, e.g. patient-related, organisational, content and format of the programme.

Table 2 Barriers: Time constraints have been reported in the literature (e.g. Ayton et al 2017), please add a comment on how it has been considered (or not) for the design of the training session and/or educational programme.

Table 2 Enablers: “Fostering patient empowerment and engagement with falls prevention” should be rephrased, as it seems to refer to the patients’ attitudes towards the falls prevention programme, or possibly towards the assistance provided by the healthcare professionals involved in the programme. Please include more details on patients’ feedback, e.g. satisfaction.

204 Please refer to Kiegaldie et al (2019), cited in Heng et al (2020), and previous work where education methods have been employed to successfully deliver falls prevention education to people with cognitive impairment. Please discuss the impact of patients’ acceptance to the programme, how to adapt the content and format to address different patients’ needs as well as the evaluation of educational outcomes.

207 Interviewees report they “felt that they were 206 not able to deliver the falls education intervention if the patient was cognitively impaired. Authors should discuss factors related to the quality of training or educational material.

255 Suggestion to split the paragraph according to the different categories of barriers presented in this section, e.g. patient-related, organisational, quality of training, context, etc.

280 From my understanding, after reading the paper, one of the main “enabler” was interviewees motivation in “fostering patient empowerment”. Is it possible that the training provided to healthcare professionals helped to create awareness on falls risk factors and preventive measures (Ayton et al 2017)?

330 Suggest re-structuring this section to highlight different categories of recommendations emerging from the analysis, e.g. patient-related, context, design of the educational content (i.e. stakeholder input), training session, organisational factors.

375 Authors mention “steps were taken to ensure effective delivery” of falls prevention intervention. Please discuss additional criteria to evaluate effectiveness, other than number of patients receiving the intervention, for a successful implementation of the programme e.g. professionals taking part in the study, feedback, patient satisfaction, etc

Reviewer 2 Report

This paper outlined the implementation of a falls prevention initiative to hospital patients and discussed the barriers and enablers that impacted program delivery by staff. I appreciate the discussion and sharing of unsuccessful implementation data, especially during the pandemic; it is important to document and demonstrate that even if interventions are 'simple' they are not always effective. 

I have some minor comments/questions for the authors, mostly geared at including additional contextual information to help the reader understand why certain decisions were made:

- why was recruitment for the follow-up study so long after the end of the RCT? I wasn't clear what the reason for the delay was 

- did members of the research team modify the Safe Recovery Program to turn it into the scripted conversation prompts? Did they have any experience in creating these types of materials? Was it tested with anyone prior to implementation?

- any thoughts on how/why the intervention group had so many more previous fallers, and were overall sicker? How do the medical and surgical wards differ, and might those differences have impacted the implementation and staff feedback? I'd recommend sharing more information about the roles of the interviewees (if possible), and how the context of the ward itself may have shaped their insights

-why were acute wards chosen for this intervention? They arguably contain more unstable/short-term patients, with higher/more stressful workloads for staff, so what was the rationale for testing the program there as opposed to the sub-acute or rehab wards referenced in the Discussion?

- it was mentioned that this was during the pandemic and PPE was involved; did masking impact program delivery at all, and might the comprehension challenges inherent between two masked individuals have been misidentified as a language/cognitive barrier?

Reviewer 3 Report

Dear authors,

This study investigated the topic of implementing patient falls education in hospitals. It makes sense for preventing patient falls and is suitable for publishing in the journal of healthcare. However, this study didn’t conduct an effective statistical analysis, and no P value was reported, which means the conclusion was not well supported by the results. This study seems more like a record, not a research paper. I have no idea whether it is possible to improve it for authors. Some small suggestions as follows:

Line 12, it should be “key to prevent”

Considering this study is a qualitative analysis, the sample size is not large enough.

The result section is mostly made up of patient interviews, I agree that the interview details are important, but they are not sufficient to support the conclusion

The conclusion is vital for preventing patient falls, but stronger evidence is needed to support it.
